# Effects of Annealing on the Properties of Gamma-Irradiated Sago Starch

**DOI:** 10.3390/molecules27154838

**Published:** 2022-07-28

**Authors:** Jau-Shya Lee, Jahurul Haque Akanda, Soon Loong Fong, Chee Kiong Siew, Ai Ling Ho

**Affiliations:** 1Faculty of Food Science and Nutrition, University Malaysia Sabah, Jalan UMS, Kota Kinabalu 88400, Sabah, Malaysia; cksiew@ums.edu.my (C.K.S.); alho@ums.edu.my (A.L.H.); 2Department of Agriculture, School of Agriculture, University of Arkansas, 1200 North University Drive, M/S 4913, Pine Bluff, AR 71601, USA; akandam@uapb.edu; 3ITS Nutriscience Sdn Bhd, 2, Jalan Sg. Kayu Ara 32/38, Berjaya Industrial Park, Shah Alam 40460, Selangor, Malaysia; slfong@its-nutriscience.com

**Keywords:** sago starch, gamma irradiation, annealing, DSC, pasting, swelling power, crystalline order, gel firmness

## Abstract

The increase in health and safety concerns regarding chemical modification in recent years has caused a growing research interest in the modification of starch by physical techniques. There has been a growing trend toward using a combination of treatments in starch modification in producing desirable functional properties to widen the application of a specific starch. In this study, a novel combination of gamma irradiation and annealing (ANN) was used to modify sago starch (*Metroxylon sagu*). The starch was subjected to gamma irradiation (5, 10, 25, 50 kGy) prior to ANN at 5 °C (T_o_-5) and 10 °C (T_o_-10) below the gelatinization temperature. Determination of amylose content, pH, carboxyl content, FTIR (Fourier Transform Infrared) intensity ratio (R_1047/1022_), swelling power and solubility, thermal behavior, pasting properties, and morphology were carried out. Annealing irradiated starch at T_o_-5 promoted more crystalline perfection as compared to T_o_-10, particularly when combined with 25 and 50 kGy, whereby a synergistic effect was observed. Dual-modified sago starch exhibited lower swelling power, improved gel firmness, and thermal stability with an intact granular structure. Results suggested the potential of gamma irradiation and annealing to induce some novel characteristics in sago starch for extended applications.

## 1. Introduction

Apart from being the most abundant carbohydrate source for human staple food, the low cost, easy tailoring, biocompatibility, renewability, and extensive sources of starch have made it a most promising biodegradable natural polymer for food and nonfood industries [1]. In its native form, starch has limited industrial application due to its low solubility, low transparency, poor heat, shear and acid stability, fast retrogradation, and poor refrigerated and frozen storage stability [2,3]. The unique molecular and granular structure of starch [3] renders it versatile for various kinds of modifications, either by enzymatic, chemical, or physical means, to achieve the desired functional properties for industrial applications. The tailored modification of starch is very essential to obtain specific functional properties for innumerous industrial applications. Even though chemical modifications provide more options for the functionalization of starch [4], the food and pharmaceutical industries prefer starches without chemical modification for safety considerations [5]. 

The ionizing radiation processing is a progressive, fast, and environmentally friendly way to produce changes in the structural and functional properties of starch [6]. The irradiation of starch also requires minimal sample preparation, does not induce a significant increase in temperature, and is free of chemical residues [7,8,9,10]. After decades of study, food irradiation was proven to be safe as a processing treatment for foods [11]. Gamma radiation alters the properties of starch by generating free radicals that break the glycoside bonds and cause the decomposition of the macromolecules, while it may also promote the cross-linking of starch molecules [8]. The advantages of ionizing radiation have attracted great research interest to investigate the effects of irradiation on various types of conventional and nonconventional starches such as arrowroot starch [12], mung bean starch [13], and kithul starch [14]. 

In recent years, there has been a growing trend toward treating starch with hydrothermal methods, including annealing. Annealing (ANN) is carried out by incubating starch samples with moisture content higher than 65% (*w*/*w*) at a temperature that is above the glass transition temperature (T_g_) but lower than the onset temperature (T_o_) for gelatinization [15]. Though simple, annealing was reported able to improve the functional properties of starch for industrial application. Annealing contributes to improving the stability of starch paste to heat and shear, increasing the paste clarity and low gelation concentration, as well as the lower digestibility in Prata banana [16]. Annealing has been interpreted as a ‘sliding diffusion’, which entails the movement of complete molecular sequences within a crystalline lattice in starch, and/or a ‘complete or partial fusion’ of crystals and the subsequent recrystallization of the melted materials at the annealing temperature [3]. The interactions between starch chains improve crystalline perfection and change the physicochemical properties of the starches. According to Zavareze and Dias [17], the extent of these changes are influenced by the starch composition and source, the ratio of amylose to amylopectin, and the treatment conditions (temperature, time, and moisture levels).

Due to the potential for the further improvement and modulation of the functional properties of starch, a combination of modification methods for starches has started to attract research attention in recent years. The combination of dual step ANN and hydroxypropylation was found to significantly improve the properties of native barley starch [18]. Hu et al. [19] reported that dual modification by enzymatic debranching and hydroxypropylation can improve the mechanical properties of normal maize starch film. Dual modification of corn starch by annealing and succinylation was found to further improve the stability of starch paste against heat and shearing stress [20]. Other starch modifications by combined treatments reported in the literature include the combination of hydroxypropylation and acid hydrolysis for potato starch [21], hydrothermal treatment with sonication for Carioca bean starch [22], the combination of ultrasound and ANN for glutinous rice [23], the combination of gamma radiation and acetylation for wheat starch [24], the combination of sonication and gamma radiation for lentil starch [25], the combination of pullulanase debranching and microwave irradiation [26], and many others. In brief, these combined modification treatments produced starches with new properties that differed from the single modification. 

Malaysia is one of the main producers of sago starch, which is extracted from the pith of a palm plant (*Metroxylon sagu*). It is considered an underutilized starch in food applications due to its poor functional properties [27]; hence, modification to improve its properties would potentially widen its utilization. In line with the ever-increasing demand for modified starch produced using a safe, fast, and economically viable method, the combination of gamma irradiation and ANN was used to modify the sago starch. To our knowledge, no work has been carried out to investigate starch modification using a combination of these two physical treatments. Our earlier preliminary work explored the effects of ANN on the pH, the degree of the short-range crystalline order, pasting, and the thermal properties of the sago starch treated with different doses of gamma radiation [28]. We found that increasing radiation doses improved the crystalline perfection by annealing, thus modifying the thermal and pasting properties of the sago starch. It is known that the extent of the crystal growth/perfection by ANN is dependent on the mobility of the glucan chains in the amorphous regions of the starch, which can be manipulated by the incubation temperature or the amount of plasticizer (water, in this case) [3]. We hypothesized that by increasing the annealing temperature, the extent of double helices formation in the irradiated starch would be further enhanced. In the previous study, the annealing temperature was at 10 °C below the onset of gelatinization. In the present study, we investigated the effect of higher annealing temperature (5 °C below the onset of gelatinization) on the irradiated sago starch and examined further the changes in the amylose content, carboxyl content, swelling power, solubility, gel firmness, and the morphology of the starch on top of the pH, the degree of short-range crystalline order, pasting, and the thermal behavior.

## 2. Results and Discussions

### 2.1. Apparent Amylose Content

The apparent amylose content of the native sago starch was found to be 31.38% (Table 1), slightly higher than the content reported by the starch manufacturer, 30.8%. This is because all the starch samples were defatted prior to the determination of the amylose content, and the removal of the fatty acids from the central hydrophobic cavity of the amylose molecules [17] made them more available to form complexes with the iodine reagent. Compared to the native sago, the amylose content of the irradiated counterparts was lower (*p* ≤ 0.05) and gradually decreased as the dose of the irradiation was increased (Table 1). Similar results were previously reported for lotus seed starch [2], brown rice starch [29], tapioca starch [7], and corn starch [30]. Increased irradiation doses induced more damage to the conformation of the amylose molecules, hence lowering the iodine-binding ability. Othman et al. [31] observed a reduction of the amylose content in gamma-irradiated sago starch at 6 kGy; however, the apparent amylose content was found increased at higher doses of 10 and 25 kGy. They attributed this to the degradation of amylopectin that released more linear chains of amylose to form a complex with iodine. This phenomenon was, however, not seen in the present study.

On the contrary, the annealed samples had significantly higher amylose content than the native starch with the ANN(T_o_-10) and showed higher amylose content than the ANN(T_o_-5) (*p* ≤ 0.05). A similar observation was reported by Babu et al. [32] for foxtail millet starch. One of the many possible alterations of the starch internal structure by annealing is the formation of double helices [17]. The more organized molecular configuration of the starch acquired by annealing may be responsible for this observation. All dual-modified starches displayed higher amylose content (*p* ≤ 0.05) than their irradiated counterparts, but insignificant differences when compared to the annealed starch samples (*p* > 0.05). Depolymerization by gamma radiation may most probably facilitate the reorganization of the starch molecules during annealing to acquire more 3-D helicoidal structures that are suitable to fix in the iodine. More amylose-like molecules were formed by the reorganization of the dual-modified starch molecules.

### 2.2. PH and Apparent Carboxyl Content

A substantial drop in pH was observed with an increase in the irradiation dose (Table 1), suggesting that oxidation had taken place with the formation of acid groups that increased the acidity of the samples. The irradiation process contributed to the oxidation of the hydroxyl groups of the starch samples, especially into the carbonyl and carboxyl groups, as well as the formation of different acids such as acetic, formic, pyruvic, and glucuronic acid [13]. This result is in corroboration with the apparent carboxyl content of the samples. Previous studies also reported similar results [7,13,14]. It is interesting to note that the pH of the dual-modified samples differed from the irradiated counterparts in a radiation dose-dependent manner: where 5kGyANN(T_o_-5) and 5kGyAnn(T_o_-10) were lower than 5kGy (*p* ≤ 0.05); insignificant differences were found between 10kGy, 10kGyANN(T_o_-5) and 10kGyAnn(T_o_-10) (*p* ≤ 0.05); whereas 25kGyANN(T_o_-5), 25kGyAnn(T_o_-10), 50kGyANN(T_o_-5), and 50kGyAnn(T_o_-10) were reported higher than 25kGy and 50kGy, respectively. The incubation temperature had no influence in this case (*p* ≤ 0.05). When compared to 25kGy and 50kGy, the dual-modified counterparts (25kGyANN(T_o_-5), 25kGyAnn(T_o_-10), 50kGyANN(T_o_-5), and 50kGyAnn(T_o_-10)) turned out to be less acidic, indicating the potential involvement of the acid groups in the structural reconfiguration of the starch macromolecules during ANN.

Since no hydrolysis of the glycosidic bonds was expected, no carboxyl content was detectable in the native and annealed samples, even though the ANN(T_o_-10) exhibited a slightly higher pH than the native counterpart (*p* ≤ 0.05). As the carboxyl content was mainly attributed to the starch degradation by irradiation, ANN did not remarkably alter the carboxyl content of the dual-modified starch, regardless of the ANN temperature. 

### 2.3. Infrared Spectra Analysis

Starch samples were analyzed by FTIR spectroscopy to confirm the breakdown of the glycosidic bonds and the changes to the short-range crystalline order (double helices) in the crystalline region and the amorphous region near the granule surface [33]. The degradation effect of gamma irradiation on the starch short-range crystalline order is seen with the increasing treatment dosage (Table 1), where the R_1047/1022_ dropped gradually until the lowest value was obtained at 50 kGy (*p* ≤ 0.05). In contrast, the R_1047/1022_ increased after ANN because the treatment provided conditions enabling the rearrangement of the starch molecules for crystalline perfection. Compared to the ANN(T_o_-10), the higher annealing temperature of ANN(T_o_-5) accelerated the rate of hydration and increased the kinetic energy of the glucan chains to take part in the molecular rearrangement. Therefore, the more pronounced effect of annealing on the structural change of the starch was obtained at a temperature closer to its gelatinization temperature. The recrystallisation-promoting effect of the higher annealing temperature of ANN(To-5) was also notable when combined with gamma irradiation. In the dual-modified samples, a synergistic effect in promoting crystalline perfection was observed, in which 25kGyANN(T_o_-5) and 50kGyANN(T_o_-5) achieved the highest R_1047/1022_ (*p* ≤ 0.05), indicating the higher molecular order of the double helix short-range in the starch granules [5]. It is postulated that these higher irradiation doses produced more new shorter molecules [30] as the starting materials to participate in the recrystallisation process, resembling the melted materials induced by ANN [3]. 

### 2.4. Swelling Power and Solubility

Upon irradiation, the swelling power of the sago starch was noticeably reduced with the increasing irradiation dose (Table 2). A reduction of almost 77.5% of the swelling ability was reported when the sago starch was irradiated up to 50 kGy. This indicates the severe depolymerization of some of the amylose and amylopectin molecules [34] and the destabilization of the hydrogen bonds within the double helices of the starch [35] under high-dose treatment. Gamma irradiation also significantly increased the amount of soluble fraction with the increasing dose. The soluble fractions are the leaching of the degraded amylose and/or amylopectin after maximum swelling; therefore, the samples that experienced more severe radiation-induced damage (contained smaller starch fractions) tended to be more soluble. The increase in the solubility was due to the increase in the polarity because of chain scission under irradiation and the decrease in interchain hydrogen bonds [36]. As expected, both of the annealed samples also showed reduced swelling power and solubility (*p* ≤ 0.05). In corroboration to the results of the FTIR, the ANN(T_o_-5), which experienced more extensive reorganization of the structure, had lower swelling power and solubility than the ANN(T_o_-10) (*p* < 0.05). Compared to gamma radiation, the suppression of solubility by ANN was more apparent than the suppression of the swelling power. According to Zavareze and Dias [17], the interplay between the extent of crystalline perfection and the amylose–amylose and/or amylose–amylopectin interactions decrease the number of available water binding sites and hence suppress the swelling of the granules and the leaching of the soluble. The swelling behavior of the dual-modified samples closely resembled the irradiated counterparts (*p* > 0.05), except for 50kGyANN(T_o_-5), which displayed the lowest swelling ability (*p* ≤ 0.05), approximately 40% lower than 50 kGy. The ANN caused a solubility suppression effect on the irradiated sago starch with a more profound outcome by ANN at (T_o_-5) than (T_o_-10). The reduction in the solubility was in proportion with the increase in the irradiation dose, where a reduction from 59% to 89% was observed for the dual-modified samples as compared to the irradiated counterparts. In brief, the dual-modified samples behaved like irradiated samples in terms of the swelling power, but the solubility was lower than the irradiated samples due to the suppression effect by ANN.

### 2.5. Gel Firmness

Table 2 shows that the native sago starch gel was able to withstand a higher deformation force than the irradiated counterparts (*p* ≤ 0.05). The formation of carboxyl groups by the radiolytic degradation of the starch molecules caused the electrostatic repulsion of the molecular association during gel formation by retrogradation [37]. As the gel is a water entrapment system by three-dimensional networks, the low intra- and intermolecular associations made the gel weaker. Gamma irradiation caused about 50% of reduction in the gel strength, independent of the radiation doses (*p* > 0.05). Depending on the starch origin and the radiation dose, the irradiated starch gel firmness/strength was found to be either improved [13] or reduced [38]. Polesi et al. [38] attributed the reduction in rice gel firmness to the excessive breakdown of the amylose molecules during irradiation, which hampered the association of molecules during retrogradation. In accordance with the reduction of the swelling power and solubility, the volume of the annealed starch gels was lowered [39] due to the lower water retention ability; hence, becoming less elastic and firmer. Concomitantly with the lowest swelling power and solubility, ANN(T_o_-5) exhibited the highest gel firmness among all tested samples (*p* ≤ 0.05). The results obtained show that ANN may improve the gel firmness of irradiated starch without being affected by the incubation temperature (*p* > 0.05). The beneficial effect was, however, only reported at lower radiation doses of 5 kGy, 10 kGy, and 25 kGy. At 50 kGy, all the irradiated samples and dual-modified samples showed insignificant differences (*p* < 0.05).

### 2.6. Thermal Properties

The gelatinization temperature of the irradiated sago starch was gradually decreased with the increasing dose of gamma radiation (Table 3), and a significant reduction was observed for T_o_ (onset temperature), T_p_ (peak temperature), and T_c_ (conclusion temperature) at 50 kGy (*p* ≤ 0.05). This observation agrees with the results of the swelling power and solubility (Table 2), whereby the depolymerization of the starch macromolecules into shorter chain molecules after irradiation weakened the associative forces in the granules and consequently eased the initiation of the phase transition and subsequent cooperative melting. Comparatively high irradiation doses, such as 50 kGy, may cause the disruption of the crystalline domain in starch granules, as well as the disruption of the double-helical order [33], hence lowering the gelatinization temperatures. Liu et al. [36] found the decreases in the gelatinization temperature and the enthalpy *(*∆*H*) of irradiated maize starch were not statistically significant from 0 to 20 kGy, but a significant decrease in T_o_, T_p_, and ∆H was observed from 20 to 50 kGy. Chung et al. [30] reported insignificant differences for the gelatinization temperature of irradiated normal corn starch from the native counterpart at 1, 5, and 10 kGy, while a significant decrease was found at 25 and 50 kGy. They related the reduced gelatinization temperature to the weaker starch granules resulting from the cleavage of glycosidic bonds by irradiation. Other inconsistent observations were also reported for the alteration of the gelatinization behavior of irradiated starch. Othman et al. [31] revealed that a small increase in the gelatinization temperature was observed for sago starch irradiated at 10 and 25 kGy. They related this observation to the presence of small molecular products (monosaccharides, small chain polysaccharides) resulting from the molecular degradation by gamma ray. According to Chung and Liu [39], gelatinization temperatures reflect the stability of starch crystallites. A decrease in the gelatinization temperature by gamma irradiation was due to the production of a defective crystalline structure and an increase in the proportion of short chains in amylopectin. On the other hand, an increase in the gelatinization temperature was attributed to the destruction of weak crystalline structures by gamma irradiation, leaving behind a more stable structure that required a higher temperature for gelatinization. In the present study, gamma irradiation did not cause any change in ∆*H* (enthalpy). The ∆*H* for irradiated wheat starch was also reported to be insignificantly different from that of the native starch until 50 kGy, which was ascribed to the unchanged crystallinity [34]. It is therefore very likely that the crystallinity of the sago starch was not severely disrupted by the irradiation condition used in the present study. 

Annealing at (T_o_-5) and (T_o_-10) brought about an increase in the T_o_ and T_p_ of the sago starch (*p* < 0.05). The increase in the gelatinization temperature by ANN has been shown to be most pronounced for T_o_ and least for T_c_ [3]. The effect of ANN on the gelatinization characteristics is well established, where there tends to be an increase in T_o_ and T_p_, a decrease in the gelatinization range, and either no change or an increase in the gelatinization enthalpy [17]. The ANN promoted bond strengthening and hence a higher temperature will be required to gelatinize the starch granules [40]. The increased ∆*H* of the annealed sago starch implies an improved starch reinforcement within the granules and the attainment of a higher configuration stability [17]. Enthalpy represents the energy needed for the dissociation of the double-helical order; therefore, the increase in the order of the double helixes (as suggested by the results of the FTIR analysis) may most probably explain the higher ∆H of the annealed starches. The granular reinforcement effect of ANN was remarkably enhanced after gamma irradiation, along with a significant raise in the gelatinization temperatures (*p* ≤ 0.05). The narrowing of the gelatinization temperature range, which may imply the higher crystallite homogeneity (double-helical structures) was also observed, particularly for ANN(T_o_-5) and the corresponding dual-modified samples. Even though the values of T_c_ for the two annealed samples are high, they are statistically insignificant from the native and irradiated samples (Table 3). This may most likely be due to the high standard deviation of 50 kGyANN(T_o_-5), 25kGyANN(T_o_-5), and 50kGyANN(T_o_-10). The high standard deviations indicate that the T_c_ values (the ending point of gelatinization) are spread out over a wider range, suggesting that these three samples contained more heterogeneous crystallites (induced by irradiation) as compared to the annealed counterparts.

Owing to the starch reinforcement by ANN, a notable increase in T_o_, T_p_, and T_c_ was depicted (*p* ≤ 0.05) in the dual-modified samples, with a more substantial change in the ANN(T_o_-5)-corresponding samples, whereby these samples appeared to have the highest gelatinization temperatures. This observation agreed well with the results obtained for the R_1047/1022_, swelling power, and solubility. By comparing the effect of ANN on high-amylose wheat (HAWS) and maize starch (HAMS) with a similar apparent amylose content, Li et al. [41] concluded that chain mobility is the key contributing factor to support greater structural rearrangement by ANN and the formation of a thermostable molecular order in the HAWS. As suggested earlier, the availability of irradiation-induced new shorter molecules and a high incubation temperature at T_o_-5 provided the required chain mobility for successive structural rearrangement in the samples. In the present study, an inconclusive change in ∆*H* was reported for all the dual-modified starches.

### 2.7. Pasting Properties

Investigation of the pasting profile plays a vital role to elucidate the technological functionalities that encompasses the gelatinization, pasting, and retrogradation characteristics of starch. The results obtained (Table 4) show that the influences of the modification treatments on the pasting temperature are in accordance with the changes seen in the gelatinization temperatures (Table 3) in the ascending order of irradiated starches < annealed starches < dual-modified starches. The enhancement of the pasting temperature by the synergistic effects of gamma radiation and ANN was more remarkable in ANN(T_o_-5)-corresponding samples.

Starch pasting involves the process of viscosity development that occurs after heating starch above the gelatinization temperature [42]. Native sago displayed the highest peak viscosity (*p* ≤ 0.05); moreover, in parallel with the effects of irradiation and ANN on the swelling capacity of the starch (Table 2), a reduction in the peak viscosities was observed for these samples. It should be emphasized that granular swelling is not the only contributing factor to the perceived paste viscosities under continuous heating and stirring during the measurement; the amylose leaching, granular rigidity and integrity, and molecular size of the irradiated-induced fractions contribute collectively to the consistency of the pastes. The change in the paste viscosity for the dual-modified samples was dose-dependent, whereby the peak viscosities for the dual-modified samples at 5 and 10 kGy were higher than the irradiated counterparts; on the other hand, the peak viscosities of the dual-modified samples at 25 and 50 kGy were lower than the irradiated counterparts. In general, the peak viscosities for the ANN(T_o_-10)-corresponding samples were reported higher than the ANN(T_o_-5)-corresponding samples (*p* < 0.05). The alteration in viscosity was in corroboration to the swelling power and solubility behaviors of these samples (Table 2). It is interesting to note that ANN may enhance the peak viscosity of the irradiated sago starch at low irradiation doses (5 and 10 kGy). Combining ANN with low irradiation doses may potentially amplify the granule rigidity and resistance to shear [43], and hence a higher viscosity. The incubation temperature for ANN also had a significant influence on the peak viscosities of the dual-modified samples, whereby at a higher incubation temperature (T_o_-5), more extensive molecular interactions took place to restrict the starch swelling and solubility, and subsequently a lower peak viscosity.

As expected, the breakdown decreased in proportion to the irradiation dose applied (*p* ≤ 0.05) because a higher irradiation dose brought about more severe molecular destruction and weaker starch granules, and hence a lower ability to withstand shearing at a high temperature. The final viscosity and setback also decreased with the increasing radiation dose. The setback and final viscosity are attributed to a reordering or polymerization of the leached amylose and long linear amylopectin [44] upon the cooling of the paste. With the increase of the radiation dose, the tendency for retrogradation was diminished, indicating the degradation of the starch macromolecules, the reduction of the crystallization capacity, and the disruption of the molecular structures [45]. The results obtained showed that the sago starch irradiated at 25 and 50 kGy contained highly fragile granules that underwent utter granular rupture with nearly negligible final viscosities and setback. Barroso and del Mastro [12] pointed out that the observed changes to the pasting properties may also be affected by the radiolytic effects of oxidation when the irradiation was carried out in the presence of air, as in the present study. The ANN brought about a decrease in the breakdown and an increase in the final viscosity of the sago starch (*p* ≤ 0.05) without affecting the setback (*p* > 0.05). The starch bond strengthening by ANN mentioned in the earlier session had improved the granular stability against collapse by continuous shearing at the maximum swelling volume. In general, all the dual-modified samples were found to exhibit a relatively high breakdown and low final viscosities and setback, indicating the irradiation exerted a more pronounced effect than ANN in the pasting properties of the dual-modified samples. 

### 2.8. Starch Morphology

Figure 1 shows the microstructure of the native and irradiated starch. Native sago granules are predominantly ovoid, with some having a spherical shape, and the presence of the typical characteristic of a truncated end [31]. Irradiation up to 50 kGy did not cause any physical damage to the sago granules. The starch granules remained smooth and intact. No significant granular size increment can be detected. Othman et al. [31] also reported that the gamma irradiation of sago starch up to 25 kGy did not affect the granule size and shape. Depending on the starch origin and irradiation dose, some researchers reported surface disruptions such as cracking, fissures, and pores [13,46]. By using gel permeation chromatography (GPC), Castanha et al. [13] found that the molecular size of mung bean starch became smaller due to the hydrolysis of the glycosidic bonds.

Dent surfaces and the development of pores on the starch granules (indicated by the arrows) are found on the ANN(T_o_-5) and the ANN(T_o_-10), as depicted in Figure 2. Xu et al. [47] reported most of the potato starch granules retained their original appearance, whereby only a small number of starch granules appeared with grooves and dents on their granule surfaces after annealing. Rocha et al. [48] found an increase of the pores on the surface of the granules for both the normal and waxy con starches and attributed this to the hydrolysis activity of the endogenous amylase under the annealing condition. No pronounced change occurred in the granule morphology when ANN was applied to green banana flour [49]. Waduge et al. [50] found that annealed starch exhibited increased granular size due to the ingress of moisture through the amorphous regions of the starch during annealing, which was not seen in this study.

Partial gelatinization had taken place in the annealed and dual-modified starches, as shown by the agglomeration of the starch granules (Figure 3 and Figure 4). No morphological differentiation can be found in these single-modified and dual-modified samples. Since the ANN temperature for the samples in Figure 3 (T_o_-5) is closer to the onset gelatinization temperature of the sago starch, more extensive agglomerations are observed compared to those annealed at (T_o_-10) (Figure 4). Differing from this study, even though Rocha et al. [48] used an ANN temperature much closer to the melting of corn starches (3 °C below T_o_), no agglomeration of the starch granules was observed by the researchers. 

## 3. Material and Methods

### 3.1. Sample Preparation

Food-grade sago (*Metroxylon sagu*) starch (12.0% moisture content, 0.12% fat content, 0.19% protein content) was bought from Nee Seng Ngeng and Sons Sago Industries Sdn. Bhd. (Malaysia). Sago starch was packed in a polyethylene bag and irradiated with a ^60^Co gamma source at the Malaysian Nuclear Agency (J.L. Shepherd Gammacell, Model 109 Irradiator). The starch samples were subjected to four doses of irradiation (5, 10, 25, 50 kGy) as described by Chung et al. [30]. The irradiation was carried out at an ambient temperature and the dose rate was 9.08 kGy/h. The dosimetry was performed using a Harwell Amber Perspex dosimeter (Type 3042 Batch H). After irradiation, annealing was performed by subjected starch slurries (starch (dry basis):water, 1:2) at 5 °C and 10 °C below the onset of gelatinization temperature in a water bath for 24 h. The onset of the gelatinization temperature of the sago starch was determined using a Differential Scanning Calorimeter (Diamond DSC, Perkin Elmer, Waltham, MA, USA) prior to the annealing. After incubation, the starch samples were centrifuged using a bench-top centrifuge (Thermo Scientific, Waltham, MA, USA) for 10 min (2000 g), decanted, washed with deionized water, and oven-dried (Memmert, Schwabach, Germany) at 40 °C to achieve a uniform moisture content (~12%). The samples were ground to pass through a sieve of 150 μm (Retsch, Haan, Germany). The samples were sealed in a polyethylene bag and kept at 4 °C until further analysis.

### 3.2. Apparent Amylose Content

The amylose content was determined using the iodine-binding spectrophotometry method modified from Williams et al. [51]. An amount of 20 mg of defatted starch (soxhlet extraction for 24 h with 95% ethanol) was added with 8 mL of 90% dimethyl sulfoxide (DMSO_4_) and mixed in a 25 mL volumetric flask. The content was heated for 15 min with continuous swirling at 85 °C until the starch sample was fully dissolved, and the content was topped up to 25 mL using distilled water. After that, 1 mL of the starch solution was added with 40 mL of distilled water and 10 mL of iodine solution, and the solution was incubated for 15 min before the measurement. The absorbance was measured (Lambda 35 UV/Vis Spectrophotometer, Perkin Elmer) at 625 nm against a reagent blank as the reference. A standard curve for mixtures of pure potato amylose and amylopectin was plotted and the regression equation of the standard curve was used to calculate the total amylose content of the samples (y = 0.210x + 0.021; R^2^ = 0.989).

### 3.3. PH and Apparent Carboxyl Content

An amount of 4 g of starch was suspended in 50 mL of distilled water and the pH value was recorded using a pH meter (Eutech pH 700, Singapore). The method of Chattopadhyay et al. [52] was used to determine the apparent carboxyl content of the starch samples. An amount of 1 g of starch was added with 25 mL of 0.1 N HCl and continuously stirred for 30 min, followed by filtration and washing with distilled water until the sample was free of chlorine. After that, 300 mL of distilled water was added, and the starch solution was boiled for 10 min to gelatinize the sample. Titration was carried out using 0.1 N NaOH with phenolphthalein as the indicator.
$$\mathrm{Apparent}\;\%\;\mathrm{carboxyl}=\frac{\left(\mathrm{sample}-\mathrm{blank}\right)\;\mathrm{ml}\;\times \mathrm{normality}\;\mathrm{of}\;\mathrm{NaOH}}{\mathrm{sample}\;\mathrm{weight}\;\left(g\;\mathrm{in}\;\mathrm{dry}\;\mathrm{basis}\right)}\times 0.045\times 100$$

### 3.4. Infrared Spectra Analysis

The infrared spectra of the starch samples were obtained on KBr pellets (starch:KBr, 1:100) of the samples using a Spectrum 100 Spectrometer (Perkin Elmer), and the intensity ratio from 1047 to 1022 cm^−1^ was calculated [33] to evaluate the starch short-range orders. Spectra were averaged over 64 scans.

### 3.5. Swelling Power and Solubility

The swelling power and solubility were determined as described by Schoch [53], with slight modification. The starch suspension (3 g of starch with 180 g of distilled water) was heated with continuous stirring at 85 °C for 30 min. After that, 20 g of distilled water was added to make up the total water to be 200 g. After centrifugation at 2200 rpm for 15 min, 50 mL of the supernatant was withdrawn, and oven dried at 105 °C for 24 h. The following calculations were employed:$$\%\;\mathrm{Solubles}\;\left(\mathrm{dry}\;\mathrm{basis}\right)=\frac{\mathrm{weight}\;\mathrm{of}\;\mathrm{soluble}\;\mathrm{starch}\;\times 400}{\mathrm{weight}\;\mathrm{of}\;\mathrm{sample}\;\mathrm{on}\;\mathrm{dry}\;\mathrm{basis}}\;\mathrm{Swelling}\;\mathrm{power}=\frac{\mathrm{weight}\;\mathrm{of}\;\mathrm{sedimented}\;\mathrm{paste}\;\times 100}{\mathrm{weight}\;\mathrm{of}\;\mathrm{sample}\;\mathrm{on}\;\mathrm{dry}\;\mathrm{basis}\;\times \left(100-\%\;\mathrm{solubles},\mathrm{dry}\;\mathrm{basis}\right)}$$

### 3.6. Thermal Properties

Thermal analysis was performed using a Differential Scanning Colorimeter (PYRIS^TM^ Diamond, Perkin Elmer) with slight modification from Chung et al. [30]. An amount of 2 mg (dry basis) of the starch sample was added with 2 mg of distilled water and kept at room temperature to equilibrate for 24 h. The sample was scanned from 30 °C to 150 °C (10 °C/min) to obtain the gelatinization parameters. Indium and tin were used as calibration standards.

### 3.7. Pasting Properties

Pasting properties of the starch were measured using a Rapid Visco Analyzer (RVA-4, Newport Scientific, Warriewood, Australia). An amount of 3.0 g of starch (dry basis) was adjusted with distilled water to achieve constant weight of 28.0 g in the aluminum canister. The heating and cooling profile used was according to Ng et al. [54]. The temperature was first held at 50 °C for 1 min and raised to 95 °C (12 °C/min) and held for another 2.5 min before cooling down to 50 °C (12 °C/min). The starch paste was held at 50 °C for 1.5 min before the pasting parameters (pasting temperature, peak viscosity, breakdown, final viscosity, and setback) were recorded.

### 3.8. Gel Firmness

Gel firmness was measured using a texture analyzer (TA.XT Plus, Stable Micro Systems, Godalming, UK) with a load cell of 25 N. The cylindrical (23 mm diameter × 20 mm height) starch gel (10% *w*/*v* d.b.) was prepared according to Adawiyah et al. [55]. An amount of 5 g of the starch sample was added with 50 mL of distilled water and cooked for 30 min (80 °C) with continuous stirring. The paste was then poured into the cylindrical mold and cooled to form a gel. The gel was stored at 4 °C for 24 h prior to the measurement. Starch gel was compressed at a speed of 1 mm/s with a compression plate (40 mm diameter) until the strain reached 90% of its original height. The maximum force recorded during compression was defined as the firmness of the gel [56].

### 3.9. Starch Morphology

The granules were observed by Scanning Electron Microscopy (EVO MA10, Carl Zeiss Microscopy, Jena, Germany) with 500× and 2000× magnification. Starch sample was suspended in a carbon tape and coated with a fin layer of gold–palladium using a sputter coater (Polaron SC500, Quorum Technologies, Lewes, UK).

### 3.10. Statistical Analysis

Data reported are the average of at least triplicate determinations (from three batches of samples prepared independently). SPSS (Statistical Package for the Social Sciences) version 20 was used for statistical analysis. One-way analysis of variance (ANOVA) with Tukey’s HSD test was used to compare the means (95% level of significance).

## 4. Conclusions

Results obtained support the hypothesis that a higher annealing (ANN) temperature (T_o_-5) enhanced the crystalline perfection by increasing the short-range orders of the irradiated sago starch as compared to a lower ANN temperature (T_o_-10). Synergism in molecular reorganization was observed when the modification was carried out by combining a high radiation dose (25 kGy and 50 kGy) with the ANN temperature of T_o_-5. Among the properties investigated, the annealing temperature significantly influenced the swelling power, the gelatinization behavior, and the pasting profile of the irradiated sago starch. The dual-modified sago starch recorded a lower swelling power and a higher thermal stability by exhibiting higher gelatinization temperatures as well as a narrower melting temperature range when compared to the native starch or the single-modified starch. The gel firmness was also significantly improved. The physical modification methods employed also offer a sago starch with new properties without destroying the granular structure, which might be important in extending its applications. Future work investigating in-depth structural changes and starch digestibility are needed to provide a better insight of the modification techniques employed.

## Figures and Tables

**Figure 1 molecules-27-04838-f001:**
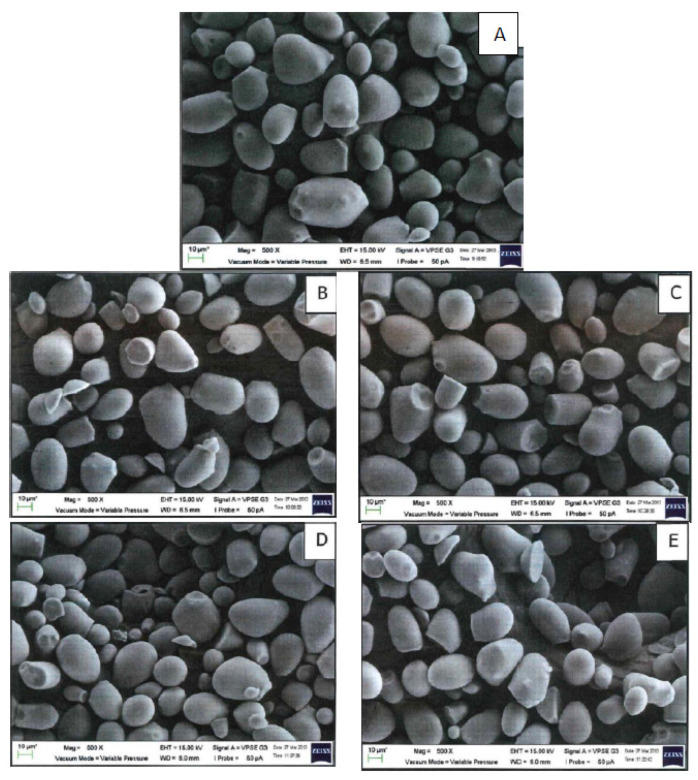
Scanning electron micrographs of (**A**) native sago; (**B**) 5 kGy; (**C**) 10 kGy; (**D**) 25 kGy; (**E**) 50 kGy (500× magnification).

**Figure 2 molecules-27-04838-f002:**
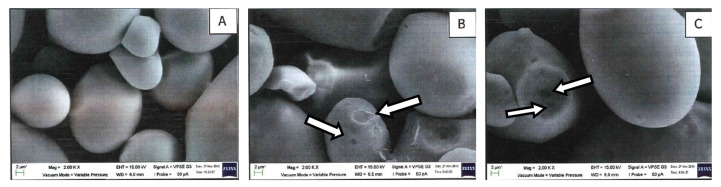
Scanning electron micrographs for (**A**) native sago; (**B**) ANN(T_o_-5); (**C**) ANN(T_o_-10) (2000× magnification).

**Figure 3 molecules-27-04838-f003:**
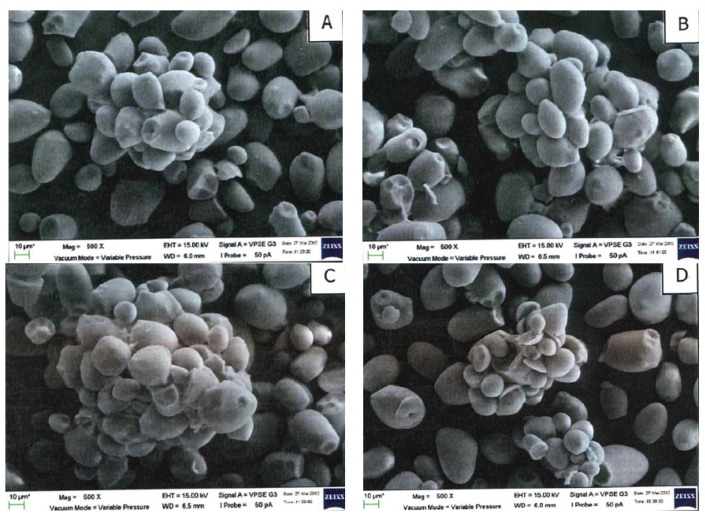
Scanning electron micrographs for (**A**) 5kGyANN(T_o_-5); (**B**) 10kGyANN(T_o_-5); (**C**) 25kGyANN(T_o_-5); (**D**) 50kGyANN(T_o_-5) (500× magnification).

**Figure 4 molecules-27-04838-f004:**
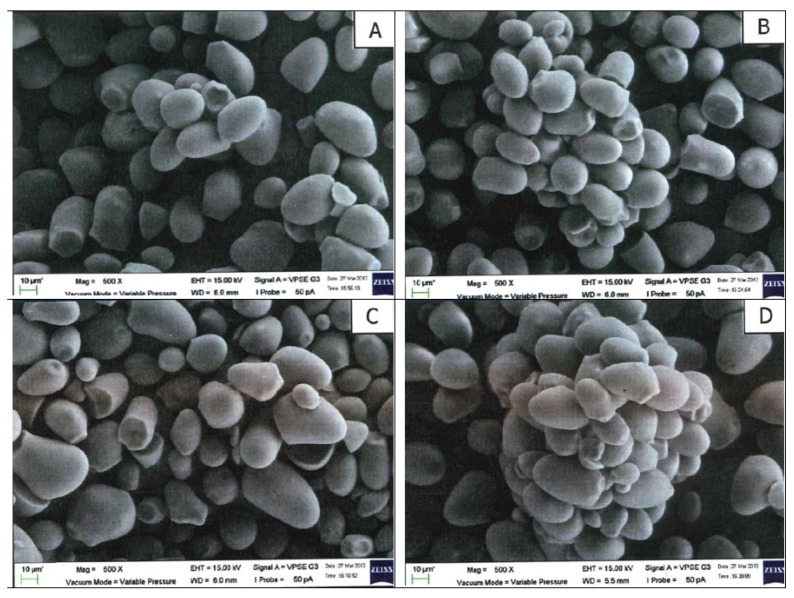
Scanning electron micrographs for (**A**) 5kGyANN(T_o_-10); (**B**) 10kGyANN(T_o_-10); (**C**) 25kGyANN(T_o_-10); (**D**) 50kGyANN(T_o_-10) (500× magnification).

**Table 1 molecules-27-04838-t001:** The apparent amylose content, pH, apparent carboxyl content, and IR intensity ratio of 1047/1022 cm^−1^ (R_1047/1022_) for native and modified sago starches.

Sample	Apparent Amylose Content (%)	pH	Apparent Carboxyl Content (%)	R_1047/1022_
Native	31.38 ± 0.13 ^c^	5.09 ± 0.02 ^h^**	N.D. *	0.57 ± 0.01 ^ef^**
5 kGy	24.21 ± 1.41 ^b^	4.91 ± 0.02 ^g^**	0.033 ± 0.006 ^a^	0.43 ± 0.01 ^c^**
10 kGy	24.45 ± 0.77 ^b^	4.12 ± 0.01 ^e^**	0.079 ± 0.002 ^b^	0.44 ± 0.02 ^c^**
25 kGy	23.64 ± 1.33 ^ab^	3.72 ± 0.01 ^c**^	0.105 ± 0.004 ^c^	0.36 ± 0.01 ^b^**
50 kGy	21.09 ± 0.19 ^a^	3.31 ± 0.01 ^a**^	0.181 ± 0.009 ^e^	0.31 ± 0.01 ^a^**
ANN(T_o_-5)	37.00 ± 0.32 ^def^	5.10 ± 0.05 ^h^	N.D. *	0.67 ± 0.00 ^i^
ANN(T_o_-10)	39.89 ± 0.33 ^g^	5.25 ± 0.02 ^i^	N.D. *	0.62 ± 0.01 ^gh^
5kGyANN(T_o_-5)	36.37 ± 0.24 ^de^	4.60 ± 0.03 ^f^	N.D. *	0.63 ± 0.01 ^h^
10kGyANN(T_o_-5)	36.12 ± 0.49 ^d^	4.14 ± 0.02 ^e^	N.D. *	0.59 ± 0.01 ^fg^
25kGyANN(T_o_-5)	37.98 ± 2.08 ^defg^	3.80 ± 0.00 ^d^	0.111 ± 0.014 ^c^	0.72 ± 0.01 ^j^
50kGyANN(T_o_-5)	36.67 ± 0.25 ^de^	3.43 ± 0.01 ^b^	0.168 ± 0.009 ^d^	0.71 ± 0.01 ^j^
5kGyANN(To-10)	38.79 ± 0.70 ^efg^	4.57 ± 0.01 ^f^**	N.D. *	0.48 ± 0.01 ^d^**
10kGyANN(To-10)	39.32 ± 0.61 ^fg^	4.17 ± 0.00 ^e^**	N.D. *	0.44 ± 0.01 ^c^**
25kGyANN(To-10)	37.37 ± 0.17 ^defg^	3.81 ± 0.01 ^d^**	0.114 ± 0.003 ^cd^	0.54 ±0.01 ^e^**
50kGyANN(To-10)	39.84 ± 1.12 ^g^	3.43 ±0.01 ^b^**	0.173 ±0.008 ^e^	0.62 ± 0.01 ^gh^**

^a–j^ Means with different lowercase letters within the same column are significantly different (*p* ≤ 0.05). * N.D. indicates ‘Not Detectable’. ** Data reported in Lee et al. [28].

**Table 2 molecules-27-04838-t002:** The swelling power and solubility (in excess water at 85 °C) and gel firmness for native and modified sago starches.

Sample	Swelling Power	Solubility (%)	Gel Firmness (g)
Native	35.92 ± 0.83 ^i^	24.50 ± 1.24 ^d^	30.67 ± 2.08 ^bc^
5 kGy	11.18 ± 0.42 ^def^	39.29 ± 0.66 ^ef^	15.33 ± 1.53 ^a^
10 kGy	10.18 ± 1.32 ^cde^	42.23 ± 2.32 ^f^	14.00 ± 1.00 ^a^
25 kGy	9.30 ± 0.38 ^cde^	65.11 ± 1.59 ^g^	15.33 ± 2.08 ^a^
50 kGy	8.09 ± 0.72 ^bcd^	85.40 ± 0.53 ^h^	15.10 ± 2.65 ^a^
ANN(T_o_-5)	17.80 ± 0.47 ^g^	8.69 ± 1.02 ^a^	68.67 ± 4.16 ^f^
ANN(T_o_-10)	24.37 ± 2.84 ^h^	15.27 ± 1.46 ^bc^	46.04 ± 3.46 ^de^
5kGyANN(T_o_-5)	13.65 ± 1.07 ^f^	16.21 ± 0.25 ^bc^	50.33 ±6.66 ^de^
10kGyANN(T_o_-5)	11.77 ± 1.57 ^ef^	16.61 ± 5.08 ^bc^	53.02 ± 0.98 ^de^
25kGyANN(T_o_-5)	8.24 ± 0.12 ^bcd^	11.21 ±1.87 ^ab^	30.33 ± 6.11 ^bc^
50kGyANN(T_o_-5)	4.86 ± 0.06 ^a^	9.55 ± 0.18 ^a^	11.35 ± 1.53 ^a^
5kGyANN(T_o_-10)	9.33 ± 0.04 ^cde^	22.10 ± 0.34 ^cd^	42.67 ± 5.51 ^cd^
10kGyANN(T_o_-10)	10.13 ± 0.17 ^cde^	43.56 ± 0.72 ^f^	60.11 ± 11.00 ^ef^
25kGyANN(T_o_-10)	7.45 ± 0.87 ^abc^	45.27 ± 3.93 ^f^	44.02 ± 4.58 ^cd^
50kGyANN(T_o_-10)	5.79 ± 0.13 ^ab^	33.76 ± 3.91 ^e^	25.10 ± 6.56 ^ab^

^a–j^ Means with different lowercase letters within the same column are significantly different (*p* ≤ 0.05).

**Table 3 molecules-27-04838-t003:** Gelatinization parameters of native and modified sago starch.

Sample	Onset Temperature, T_o_ (°C)	Peak Temperature, T_p_ (°C)	Conclusion Temperature, T_c_ (°C)	Melting Temperature Range, ∆T (°C)	Enthalpy ∆*H* (J/g)
Native **	69.06 ± 0.02 ^ab^	74.69 ± 0.39 ^b^	79.96 ± 0.41 ^abc^	10.90 ± 0.43 ^ab^	5.43 ± 0.73 ^ab^
5 kGy **	69.39 ± 0.33 ^b^	74.08 ± 0.39 ^b^	78.80 ± 0.93 ^ab^	9.41± 0.91 ^ab^	4.54 ± 1.29 ^ab^
10 kGy **	69.50 ± 0.51 ^b^	74.17 ± 0.18 ^b^	78.39 ± 0.20 ^ab^	8.89 ± 0.57 ^ab^	4.21 ± 0.61 ^ab^
25 kGy **	69.23 ± 0.19 ^b^	73.99 ± 0.17 ^b^	78.78 ± 0.40 ^ab^	9.55± 0.20 ^ab^	6.20 ± 0.91 ^ab^
50 kGy **	66.12 ± 0.47 ^a^	72.44 ± 0.10 ^a^	77.03 ± 0.35 ^a^	10.91± 0.24 ^ab^	7.75 ± 1.96 ^ab^
ANN(T_o_-5)	75.81 ± 0.16 ^d^	77.88 ± 0.17 ^d^	81.86 ± 0.26 ^abcd^	6.05 ±0.12 ^a^	10.29 ± 0.69 ^b^
ANN(T_o_-10)	72.22 ±0.27 ^c^	75.57 ± 0.42 ^c^	80.54 ± 0.90 ^abc^	8.32 ± 0.63 ^ab^	10.15 ± 1.77 ^b^
5kGyANN(T_o_-5)	79.49 ± 0.14 ^f^	81.25 ± 0.17 ^gh^	85.22 ± 0.35 ^cdef^	5.73 ±0.22 ^a^	7.17 ± 2.30 ^ab^
10kGyANN(T_o_-5)	80.02 ± 0.06 ^fg^	81.98 ± 0.19 ^h^	86.82 ± 0.73 ^def^	6.80 ± 0.70 ^ab^	8.23 ± 2.46 ^ab^
25kGyANN(T_o_-5)	80.64 ± 0.41 ^g^	83.06 ± 0.20 ^i^	88.57 ± 0.54 ^ef^	7.93 ± 0.94 ^ab^	6.21 ± 4.17 ^ab^
50kGyANN(T_o_-5)	80.50 ± 0.50 ^fg^	83.74 ± 0.10 ^i^	89.41 ± 2.84 ^f^	8.91 ± 3.33 ^ab^	3.18 ± 4.26 ^ab^
5kGyANN(T_o_-10) **	76.07 ± 0.04 ^d^	78.32 ± 0.10 ^d^	81.99 ± 0.10 ^abcd^	5.92 ± 0.08 ^a^	5.85 ± 2.16 ^ab^
10kGyANN(T_o_-10) **	76.68 ±0.51 ^de^	79.38 ± 0.29 ^e^	83.61 ± 0.74 ^bcde^	6.93 ± 0.09 ^ab^	6.03 ± 3.77 ^ab^
25kGyANN(T_o_-10) **	77.34 ± 0.55 ^e^	80.39 ± 0.34 ^f^	89.61 ± 5.96 ^f^	12.27 ± 5.60 ^b^	6.35 ± 4.81 ^ab^
50kGyANN(T_o_-10) **	77.75 ± 0.65 ^e^	81.02 ± 0.35 ^fg^	86.53 ± 4.45 ^def^	8.78 ± 1.88 ^ab^	8.22 ± 1.26 ^ab^

^a–i^ Means with different lowercase letters within the same column are significantly different (*p* ≤ 0.05). ** Data for these samples are adopted from Lee et al. [28].

**Table 4 molecules-27-04838-t004:** Pasting properties of native and modified sago starch.

Sample	Pasting Temperature (°C)	Peak Viscosity (cP)	Breakdown (cP)	Final Viscosity (cP)	Setback (cP)
Native **	74.38 ± 0.08 ^c^	397.92 ± 2.17 ^n^	256.17 ± 1.64 ^k^	193.14 ± 4.51 ^h^	51.39 ± 4.84 ^f^
5 kGy **	73.88 ± 0.20 ^b^	179.00 ± 2.49 ^g^	131.42 ± 0.55 ^e^	60.64 ± 1.48 ^e^	13.05 ± 0.86 ^c^
10 kGy **	73.60 ± 0.05 ^b^	171.69 ± 0.79 ^f^	158.89 ± 0.85 ^f^	16.78 ± 0.90 ^c^	3.97 ± 0.57 ^ab^
25 kGy **	73.63 ± 0.03 ^b^	128.14 ± 0.56 ^e^	128.06 ± 0.70 ^e^	2.39 ± 0.26 ^b^	2.31 ± 0.32 ^ab^
50 kGy **	72.37 ± 0.03 ^a^	53.77 ± 0.42 ^c^	54.89 ± 0.54 ^c^	0.29 ± 0.04 ^a^	1.42 ± 0.09 ^ab^
ANN(T_o_-5)	77.85 ± 0.26 ^e^	387.83 ± 3.03 ^m^	167.08 ± 3.75 ^g^	274.72 ± 1.61 ^j^	53.97 ±0.80 ^f^
ANN(T_o_-10)	75.87 ± 0.28 ^d^	370.33 ± 6.96 ^l^	198.42 ± 3.56 ^i^	224.86 ± 3.76 ^i^	52.94 ± 0.34 ^f^
5kGyANN(T_o_-5)	80.73 ± 0.03 ^i^	243.60 ± 0.80 ^j^	240.80 ± 0.90 ^j^	91.88 ± 0.84 ^g^	89.08 ± 0.77 ^g^
10kGyANN(T_o_-5)	81.55 ± 0.05 ^j^	191.81 ± 1.40 ^h^	193.81 ± 1.25 ^h^	26.29 ± 0.07 ^d^	28.29 ± 0.18 ^e^
25kGyANN(T_o_-5)	82.78 ± 0.03 ^k^	99.44 ± 0.21 ^d^	102.11 ± 0.21 ^d^	2.44 ± 0.42 ^b^	5.10 ± 0.41 ^b^
50kGyANN(T_o_-5)	82.67 ± 0.08 ^k^	21.78 ± 0.68 ^a^	22.06 ± 0.10 ^a^	0.52 ± 0.45 ^a^	0.79 ± 1.09 ^a^
5kGyANN(T_o_-10) **	77.93 ± 0.03 ^e^	251.65 ± 1.40 ^k^	193.45 ± 0.40 ^h^	78.43 ± 0.25 ^f^	20.23 ± 1.44 ^d^
10kGyANN(T_o_-10) **	78.70 ± 0.05 ^f^	207.91 ± 0.09 ^i^	189.55 ± 0.04 ^h^	22.41 ± 0.36 ^d^	4.04 ± 0.41 ^ab^
25kGyANN(T_o_-10) **	79.57 ± 0.03 ^g^	127.39 ± 0.25 ^e^	126.89 ± 0.25 ^d^	2.53 ± 0.46 ^b^	2.03 ± 0.46 ^ab^
50kGyANN(T_o_-10) **	79.93 ± 0.03 ^h^	34.34 ± 0.86 ^b^	35.39 ± 0.97 ^b^	0.22 ± 0.08 ^a^	1.12 ± 0.11 ^ab^

^a–m^ Means with different lowercase letters within the same column are significantly different (*p* ≤ 0.05). ** Data for these samples are adopted from Lee et al. [28].

## Data Availability

Not applicable.

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
