# Peer review of "Effects of Annealing on the Properties of Gamma-Irradiated Sago Starch"

_molecules, 2022, doi:10.3390/molecules27154838_

Round 1

Reviewer 1 Report

1.The abstract did not write valuable research results and conclusions.

2.The data lacks correlation analysis and is only the result description.

3.The experimental method is not detailed. This seriously affects the credibility and repeatability of the manuscript.

4.Some formulas have obvious errors.

Reviewer 2 Report

In my opinion manuscript molecules-1789370 is well written and deserves publication after revision.

Some suggestion to improve the manuscript:

1.       Highlight novelty at the end of the abstract.

2.       Check English especially in the introduction: line 37 versatility should be versatile.

3.       Emphasize originality at the end of the introduction.

4.       Figures 1 - 4 give scale bars and draw arrows (line 367)

5.       Indicate dose rate, dose confidence interval. What dosimeters was used in gamma irradiation experiments?

6.       Give ftir and dsc data in supplementary materials

Reviewer 3 Report

In this manuscript the authors made an interesting study and this investigation can be an important contribution to the scientific literature. The authors investigated the effect of combination of annealing modification at two different temperatures and irradiation on sago starch properties. The study was well designed and the manuscript was well prepared, the paper is written clearly, and the results are sufficiently interpreted and discussed with citing others' research. The methodologies are appropriate to elucidate the questions to be answered. I only have some specific suggestions to improve the scientific value.

Lines 121- 178.  Are the FTIR spectra visible changes only at 1022 cm-1 and 1047 cm-1 for modified samples? Please, add FTIR spectra of all or selected samples to the manuscript and discuss them.

Round 2

Reviewer 1 Report

The author made detailed modifications. I think it has reached the publishing level.